# Across Dimensions: Developing 2D and 3D Human iPSC-Based Models of Fragile X Syndrome

**DOI:** 10.3390/cells11111725

**Published:** 2022-05-24

**Authors:** Azalea Lee, Jie Xu, Zhexing Wen, Peng Jin

**Affiliations:** 1Neuroscience Graduate Program, Emory University, Atlanta, GA 30322, USA; azalea.lee@emory.edu; 2MD/PhD Program, Emory University School of Medicine, Atlanta, GA 30322, USA; 3Genetics and Molecular Biology Graduate Program, Emory University, Atlanta, GA 30322, USA; jie.xu@emory.edu; 4Department of Psychiatry and Behavioral Sciences, Emory University School of Medicine, Atlanta, GA 30322, USA; 5Department of Cell Biology, Emory University School of Medicine, Atlanta, GA 30322, USA; 6Department of Human Genetics, Emory University School of Medicine, Atlanta, GA 30322, USA

**Keywords:** fragile X syndrome, FMRP, iPSC, organoids

## Abstract

Fragile X syndrome (FXS) is the most common inherited cause of intellectual disability and autism spectrum disorder. FXS is caused by a cytosine-guanine-guanine (CGG) trinucleotide repeat expansion in the untranslated region of the *FMR1* gene leading to the functional loss of the gene’s protein product FMRP. Various animal models of FXS have provided substantial knowledge about the disorder. However, critical limitations exist in replicating the pathophysiological mechanisms. Human induced pluripotent stem cells (hiPSCs) provide a unique means of studying the features and processes of both normal and abnormal human neurodevelopment in large sample quantities in a controlled setting. Human iPSC-based models of FXS have offered a better understanding of FXS pathophysiology specific to humans. This review summarizes studies that have used hiPSC-based two-dimensional cellular models of FXS to reproduce the pathology, examine altered gene expression and translation, determine the functions and targets of FMRP, characterize the neurodevelopmental phenotypes and electrophysiological features, and, finally, to reactivate *FMR1*. We also provide an overview of the most recent studies using three-dimensional human brain organoids of FXS and end with a discussion of current limitations and future directions for FXS research using hiPSCs.

## 1. Introduction

### 1.1. Fragile X Syndrome 

Fragile X syndrome (FXS) is a neurodevelopmental disorder characterized by developmental delay, intellectual disability, autism, attention-deficit/hyperactivity disorder, anxiety, and seizures [1,2,3]. FXS is the most common inherited cause of intellectual disability and the leading monogenic cause of autism spectrum disorder [1,3,4,5]. The study of FXS has provided, and will continue to provide, crucial insight to understanding the pathophysiology of the disorders and symptoms it encompasses. FXS is caused by an expansion of cytosine-guanine-guanine (CGG) trinucleotide repeats in the 5′-untranslated region (UTR) of the fragile X messenger ribonucleoprotein 1 (*FMR1*) gene on the X chromosome. When an individual has a full mutation with greater than 200 CGG repeats, the repeats themselves, as well as the promoter region of the *FMR1* gene, are hypermethylated [6,7,8,9]. Due to abnormal methylation and histone modifications [10], *FMR1* is silenced, leading to subsequent reduction or absence of its protein product fragile X messenger ribonucleoprotein (FMRP) [2,3,9,10,11,12]. FMRP is an RNA-binding protein that can modulate the translation of its mRNA targets involved in diverse essential cellular functions, including neurogenesis, axon and dendrite formation, synaptic plasticity, cytoskeletal organization, histone modification, and RNA transport [12,13,14,15]. The loss of FMRP has many consequences, with the clinical manifestation being FXS. Individuals with repeat numbers between 55 and 200 are referred to as premutation carriers and may develop other *FMR1*-associated disorders, such as fragile-X-associated tremor ataxia syndrome (FXTAS) for males and some females and fragile-X-associated primary ovarian insufficiency (FXPOI) for females [1,3,16,17,18].

Thirty years have passed since the discovery of the genetic and molecular mechanism of FXS [7,8,19], which has paved the way for understanding disorders similarly caused by nucleotide repeat expansions in other genes, including Huntington’s disease, Friedreich’s ataxia, myotonic dystrophy, spinocerebellar ataxias, and familial adult myoclonic epilepsies [20]. Active research on FXS has led to substantial progress in understanding the pathogenesis of the disorder. Nonetheless, there is not yet a gold standard treatment available for FXS. Various promising therapeutic targets for FXS have been discovered and tested in animal models of FXS, but most clinical trials have been unsuccessful [21,22,23,24]. Although animal models of FXS [25,26], including *Drosophila*, mouse, rat, and zebrafish, are invaluable in understanding the functions of FMRP, there are limitations because the pathogenic mechanism of FXS in humans, in which trinucleotide repeat expansions lead to gene silencing, does not apply to other species [6,12,27,28,29]. Knock-in mouse models with expanded CGG repeats do not reproduce the pathophysiological processes of FXS in humans [30,31]. Furthermore, recent studies have discovered that FMRP targets are species-specific: FMRP in humans may bind different RNAs than does FMRP in animal models of FXS [22,24]. Different neural cell types and brain organoids derived from human induced pluripotent stem cells (iPSCs) provide a unique means of studying the normal and pathological processes of neurodevelopment seen in FXS. In addition, using iPSC-derived cellular models of FXS enables the manipulation of various genes or proteins that are implicated to be altered in FXS. Examining the effects of these modulations provides a deeper insight into the pathogenesis, as well as highlighting promising therapeutic targets for FXS.

### 1.2. Pluripotent Stem Cells (PSCs) and Translational Applications

The advent of stem cell technology has opened a new avenue for studying human development and diseases in vitro. Unlike human postmortem tissues that are limited in quantity and represent only the end-stage status of a disease or human primary cells that can only be maintained for a limited time in vitro, human pluripotent stem cells (hPSCs), with the ability to self-renew indefinitely and the potential to differentiate into almost every cell type in the body, provide a renewable and inexhaustible resource for studying human development and diseases, as well as for testing therapeutic compounds and other treatment strategies.

There are two types of human pluripotent stem cells: human embryonic stem cells (hESCs) and human induced pluripotent stem cells (hiPSCs). hESCs are obtained primarily from the inner cell mass of supernumerary blastocysts created by in vitro fertilization (IVF), a method of assisted reproduction [32]. These pluripotent stem cells can be differentiated into various cell types of interest and used to investigate numerous processes of development and disease mechanisms. Many studies have used hESC-derived neural cells as a model for studying fragile X syndrome [28,29,33,34,35,36,37,38]. For example, Telias et al. (2013) investigated three male hESC lines (FXS-hESCs) that were obtained from spare IVF-derived embryos diagnosed with fragile X syndrome via preimplantation genetic diagnosis (PGD) [34]. They found that *FMR1* was expressed in undifferentiated FXS-hESCs and was progressively inactivated through differentiation into neurons, consistent with the process that occurs in human FXS fetuses. Moreover, they observed aberrant expression patterns of key neural genes and delayed development of neural rosettes in FXS lines during early stages of differentiation, as well a significant shift towards the glial lineage during final stages of differentiation, leading to poor neuronal maturation and high gliogenic development in FXS lines [34].

Although such FXS-hESCs are advantageous in terms of recapitulating the in vivo process of *FMR1* inactivation during differentiation, they are difficult to acquire given the limited resource of IVF-derived pre-diagnosed embryos. Therefore, an alternative approach has been taken utilizing gene editing techniques to create knockout (KO) of the *FMR1* gene in healthy control hESC lines to model FXS in vitro. As an example, Li et al. (2020) made use of clustered regularly interspaced short palindromic repeats (CRISPR) technique to generate *FMR1*-KO hPSC lines (two *FMR1*-KO hESC lines and one *FMR1*-KO hiPSC line) for studying FMRP function [23]. Nevertheless, the use of hESCs raises ethical and legal issues and, therefore, is under strict regulations in many countries [39].

Human iPSCs, the other type of human pluripotent stem cells, have also been widely used for disease modeling since its first establishment in 2007 [40]. hiPSCs are derived from human somatic cells via reprograming with four transcription factors (Oct3/4, Sox2, Klf4, and c-Myc, also called Yamanaka factors) and are, therefore, more accessible than hESCs [40,41]. Since they carry the exact same genetic information as the donor, patient-derived hiPSCs would naturally contain the pathogenetic mutations and are, therefore, extremely helpful when differentiating into cell types of interest as models for studying disease mechanisms. Once a patient iPSC line(s) is established, it is also possible to perform large-scale high-throughput drug screening to identify effective drug treatments for the disease of interest. Furthermore, patient-specific iPSCs have promoted the development of regenerative medicine and cell therapies in a way that makes genetically corrected autologous cell transplantation possible and eliminates the possibilities of immune rejection from allogeneic cell transplantation [42,43,44]. Indeed, numerous studies have taken advantage of such disease models for their investigations.

Here in this review, we focus on compiling and summarizing research findings on hiPSC-based FXS models (Table 1; studies listed by section in the order of discussion). We also discuss the future directions of hiPSC-based studies on FXS.

## 2. Two-Dimensional (2D) Human iPSC-Based Cellular Models of FXS

Two-dimensional cellular models have several advantages, including the ability to culture large quantities of a relatively homogenous population of cells and maintain those cells at a particular stage of development. Using 2D cell models of disease, cell-type-specific phenotypes can be easily examined. Moreover, cells at certain developmental stages can be monitored and followed through stages of maturation and differentiation in a phase-locked fashion [45]. This section provides a review of the studies that have utilized two-dimensional iPSC-based cellular models of FXS and their findings.

### 2.1. Generation of Human iPSC-Derived Cellular Models of FXS 

Early studies that generated iPSCs derived from primary cells of patients with FXS (hereafter referred to as FXS-iPSCs) found that *FMR1* was inactive in iPSCs and its expression was not reactivated by reprograming the original somatic cells into iPSCs [46,47], nor by differentiation into neurons [47] (Figure 1A). *FMR1* was confirmed to remain silent in iPSCs derived from dermal fibroblasts from a pediatric FXS patient, those from an adult patient of FXS, as well as lung fibroblasts from a fetus diagnosed with FXS [46]. The *FMR1* locus was methylated in FXS-iPSCs and contained histone modifications associated with a heterochromatin state of epigenetic markers, including increased H3K9 methylation and reduced H3 acetylation and H3K4 methylation [46]. The absence of FMRP expression further verified the silencing of *FMR1* in FXS-iPSCs [46,47,48].

While most molecular mechanisms of FXS are maintained in iPSC-based models of the disorder, a few studies have reported discrepancies in primary cells and iPSCs with fragile X full mutations. One study, which examined the level of 5-hydroxymethylcytosine (5hmC), an intermediate epigenetic marker for DNA demethylation, at the *FMR1* promoter region, reported differential findings in primary neurons of FXS patients and iPSCs derived from FXS patients [49]. 5hmC was found to be enriched at the *FMR1* promoter in primary neurons isolated from postmortem brain tissues of FXS patients compared to controls but not in FXS-iPSCs, FXS-iPSC-derived neurons, or non-neuronal cell lines collected from FXS patients [49].

Another study suggested that the reprograming process could introduce instability of the expanded CGG repeats in that the iPSCs generated from fibroblasts of FXS patients had variable ranges of CGG repeat lengths, although all above the normal repeat size, that could differ from those of the original fibroblast population [47]. Furthermore, *FMR1* was found to be methylated in subsets of iPSCs generated from fibroblasts of individuals with an unmethylated full mutation (UFM), supporting that the reprograming process can direct silencing of *FMR1* with a full mutation [50,51]. The iPSC lines from UFM individuals were found to have a higher CGG repeat size threshold for *FMR1* silencing—greater than 400—compared to the threshold of 200 repeats seen in FXS patients [51]. These findings emphasize the importance of determining CGG repeat lengths and *FMR1* silencing status in any FXS-iPSC cell line.

**Figure 1 cells-11-01725-f001:**
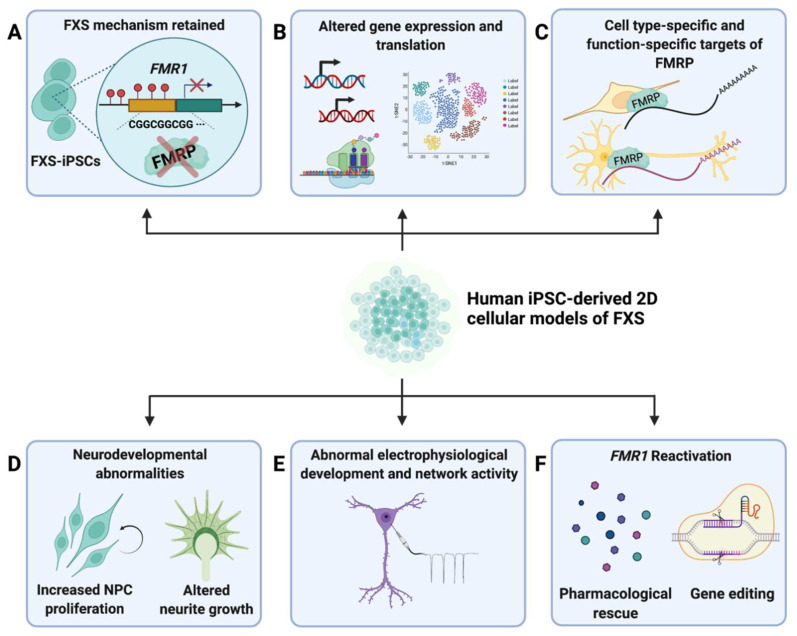
**Summary of major findings from 2D FXS−iPSC−derived models.** (**A**) Studies that have utilized 2D hiPSC-derived models of FXS have found that the fundamental molecular mechanisms of FXS pathology is retained in iPSCs derived from primary cells of FXS patients (FXS-iPSCs) [46,47]. (**B**) FXS-iPSCs and derived neural cells showed altered gene expression and translation compared to controls [21,52,53,54,55,56,57]. (**C**) Studies of targets and functions of FMRP showed that there are cell-type-specific [23] and function-specific [58] targets of FMRP [23,58]. (**D**) Neurodevelopmental abnormalities, including increased NPC proliferation [21,23,53] and altered neurite growth [47,48,53,55,59], were characterized in iPSC-derived models of FXS. (**E**) FXS-iPSC-derived neural cells exhibited electrophysiological abnormalities [24,59,60,61,62,63]. (**F**) Pharmacological rescue [64,65,66,67,68,69] and gene editing [70,71,72,73] are two major methods of *FMR1* reactivation that have been explored.

### 2.2. Gene Expression and Translation in the 2D hiPSC-Derived Cell Models of FXS

Following the generation of FXS-iPSCs and confirmation that the molecular features of the disorder are conserved in these cells, several studies have used FXS-iPSCs to reveal altered gene expression in FXS (Figure 1B). A DNA microarray analysis by Halevy et al. (2015) and a more recent study using RNA sequencing by Utami et al. (2020) both showed that several genes associated with neuronal differentiation, neuronal projection, axogenesis, axon guidance, and synaptic signaling were downregulated in neurons derived from FXS-iPSCs compared to controls [52,53]. These genes included *ROBO3*, *SLIT1*, *DCC*, *srGAP3*, and *CRMP*. Many downregulated genes in FXS-iPSC-derived neurons were found to bind the repressor element 1 silencing transcription factor (REST), which inhibits neural gene expression in non-neuronal tissues and decreases in expression with the progression of neural differentiation [52,74]. To further elucidate the interaction between FMRP and REST, Halevy et al. (2015) conducted a microRNA (miRNA) array analysis and identified hsa-mir-382, which bound REST and was downregulated in FXS-iPSC-derived neurons. Overexpression of hsa-mir-382 led to an increase in levels of REST target genes involved in axon guidance that were found to be downregulated in FXS-iPSC-derived neurons. This finding indicates that, when FMRP is absent, hsa-mir-382 levels are decreased, leading to decreased repression of REST, which results in a stronger suppression of neural genes essential for normal differentiation and development [52]. Another study using RNA sequencing by Lu et al. (2016) found hundreds of differentially expressed genes (DEGs) in FXS-iPSCs and iPSC-derived neurons compared to control iPSCs and iPSC-derived neurons [54]. Most genes were related to neural development, potassium channels, and glutamatergic synapses. Genes involved in neuronal differentiation and neural development that were upregulated included *WNT1*, *BMP4*, *POU3F4*, *TFAP2C*, and *PAX3*, some of which are known to be involved in deterring cells from neuronal differentiation and supporting the maintenance of undifferentiated proliferative states. In contrast, many genes coding for potassium channels, including *KCNA1, KCNC3, KCNG2*, *KCNIP4*, *KCNJ3*, *KCNK9*, and *KCNT1,* were downregulated [54]. Boland et al. (2017) confirmed the dysregulated expression of *ROBO*, *SLIT*, and another REST target gene, *TSPAN7,* in neural progenitor cells (NPCs) and neurons derived from FXS-iPSCs [55]. In addition to an altered expression of many genes involved in neuronal development revealed in earlier studies, the authors found differentially expressed genes associated with developmental signaling and cell migration (*BAMBI*, *BMP7*, *DKK1*, and *CHCHD2*) in FXS-iPSC-derived neural cells compared to controls [55].

In line with findings of altered expression of genes involved in neuronal differentiation [52,54,55] in iPSC models of FXS, several studies have revealed an imbalance in neuronal–glial development. Two studies that used NPCs derived from *FMR1*-knockout human iPSCs created using the CRISPR/Cas9 system found that the *FMR1*-null NPCs had elevated expression of glial fibrillary acidic protein (GFAP), a marker for astrocytes [24,56]. Although these cells appeared to have a morphology similar to glial cells, immunofluorescence staining of GFAP-positive cells showed that these cells were positive for SOX2 and NESTIN, both of which are markers for NPCs [56]. The group concluded that these are NPCs with an abnormal morphology and glial gene expression rather than true astrocytes. Interestingly, the elevated GFAP expression was reversible through reintroduction of FMRP into *FMR1*-null NPCs via a lentiviral vector, and the reduction in GFAP levels was found to be proportional to the increase in expression of FMRP [56].

Recently, a group developed a flow-cytometry-based high-throughput single-cell assay that measures translation and proliferation markers specific to the type of neural cell [21]. Global protein synthesis was elevated in FXS-iPSC-derived NPCs compared to controls (Figure 1B). Moreover, the FXS NPCs appeared to more strongly express markers of proliferation than those of neurogenesis. The translational defects were more obvious in early proliferative cells than more mature cells [21,53]. The group further explored the dysregulated protein synthesis seen in FXS NPCs by investigating the phosphoinositide 3-kinase (PI3K) pathway, which has been found to be overactive in FXS by previous research [75,76,77]. An inhibitor of p110β, a catalytic subunit known to play a key role in driving the hyperactive signaling of the PI3K pathway in FXS, was able to ameliorate the protein synthesis defects in FXS NPCs [21].

A recent study explored a possible mechanism that could explain the dysregulation of genes in FXS. Kurosaki et al. (2021) found that the nonsense-mediated mRNA decay (NMD) pathway modulates the expression of genes via interaction with FMRP [57]. A key NMD factor UPF1 was found to bind directly to FMRP and promote the binding of FMRP to targets of the NMD pathway. FMRP normally acts to suppress NMD, which means NMD is hyperactivated in FXS. Three NMD inhibitors tested in the study (NMDI-1, NMDI-14, and curcumin) were able to partially rescue the expression of neuronal markers and increase neurite growth in FXS-iPSC-derived neurons, although not to the level seen in control iPSC-derived neurons [57]. Of note, however, is that NMD hyperactivation is present in FXS-iPSCs and primary neural cells but absent in neurons derived from these iPSCs, which implies that the process is regulated by other cellular and molecular factors. One example of another mRNA degradation pathway that may play a role in FXS is the N^6^-methyladenosine (m^6^A) pathway, which is the most prevalent internal modification mechanism of eukaryotic mRNAs [78,79,80]. One study found that FMRP targets were enriched with m^6^A markers and FMRP interacts with YTH N^6^-methyladenosine RNA binding protein 2 (YTHDF2) [79], one of the m^6^A readers [79,80].

Several studies, as summarized above, have revealed differential gene expression in FXS-iPSCs and iPSC-derived neural cells. Few have investigated potential mechanisms for the altered transcription. Future studies examining the function of FMRP in dysregulated gene expression would further our understanding of the mechanisms of FXS pathology.

### 2.3. Studies of FMRP Function and Targets

To identify RNA targets bound by FMRP, Li et al. (2020) applied (1) high-throughput sequencing of RNA isolated by crosslinking immunoprecipitation (HITS-CLIP or CLIP-seq) using dorsal and ventral forebrain NPCS and neurons derived from FMRP-tagged hPSCs (both iPSC and ESC lines) and (2) RNA sequencing of the same four cell types derived from isogenic *FMR1* knockout hPSC lines [23]. Consistent with other studies of gene expression and transcription of FXS-iPSC-derived neural cells [52,53], the group’s transcriptome analysis revealed downregulated genes associated with neuronal differentiation and synapse formation in *FMR1*-KO dorsal NPCs. KO NPCs showed a higher level of proliferation and lower level of differentiation than control NPCs [23], as found in other studies previously discussed (Figure 1D) [21,53]. Interestingly, however, there was little overlap between DEGs identified by RNA-seq and *FMR1* targets identified by CLIP-seq in each cell type, implying that targets of FMRP are distinct from DEGs in FXS. The authors state that their findings emphasize that FMRP regulates many cellular functions at multiple levels. FMRP targets common to all four types of cells, as well as cell-type-specific targets, were identified. Among the identified type- or location-specific targets of FMRP was *PIK3CB* in dorsal NPCs and neurons, highlighting the involvement of FMRP in the PI3K pathway [21,75,76,77,81,82]. In ventral neurons, *KIF3B*, known to be crucial in axon development [83], and *CTNNB1*, found to modulate ribosome levels, were among the top hits as FMRP targets. Of note, *CTNNB1* was found to be downregulated in KO ventral neurons but upregulated in dorsal NPCs, while no difference between KO and controls was seen in other cell types, meaning that the regulation by FMRP is cell-type-specific (Figure 1C). The study also found that, overall, FMRP bound preferentially to longer RNA targets and to coding regions of mRNAs in contrast to 5′ and 3′ UTRs of mice [13]. In NPCs and ventral neurons, a weak preference for mRNAs with G-quadruplex structures was found, in line with several previous studies [84,85,86].

Another recent study examined the role of FMRP on RNA localization by combining techniques of mechanical subcellular fractionation (soma and neurite separation) and RNA sequencing in both FXS-iPSC-derived neurons and FMRP-KO mouse CAD neuronal cells [58]. Among previously identified FMRP targets in three CLIP-seq studies [13,18,87], the authors found a subset of genes that exhibited decreased neurite localization in FXS cell lines and termed them FMRP localization targets. In both human and mouse cell lines, FMRP localization target genes contained more G-quadruplex structures in their 3′ UTRs than genes that were not targets of FMRP. Interestingly, ribosome profiling analysis of FMRP targets in mouse CAD cells showed that this enrichment of G-quadruplexes was absent in FMRP targets of translational regulation, partially explaining the diverging results on FMRP binding motifs in previous research [84,85,86,88]. This finding indicates that there are distinct patterns of FMRP binding to its targets of translational regulation compared to those of subcellular localization (Figure 1C). Moreover, although not tested in hiPSC-derived neurons, the FMRP I304N mutant with dysfunctional translational regulation [89] displayed intact localization of target transcripts in mouse CAD cells, implying two independent functions by FMRP. Further research of FMRP’s functions and targets in humans is necessary.
cells-11-01725-t001_Table 1Table 1Summary of relevant findings from primary literature reviewed in text organized by section.ReferenceSection(s) of Review Model Type(s)Summary of FXS-iPSC-Relevant FindingsUrbach et al. (2010) [46]***Generation of human iPSC-derived cellular models of FXS***Characterization of FXS-iPSCsFXS-iPSCsIn FXS-iPSCs,*FMR1* was inactive.*FMR1* hypermethylation was present.*FMR1* contained histone modifications associated with a heterochromatin state.FMRP expression was absent.Sheridan et al. (2011) [47]***Generation of human iPSC-derived cellular models of FXS***Characterization of FXS-iPSCs***Characterization of neurodevelopmental and electrophysiological abnormalities of iPSC-derived cell models of FXS***FXS-iPSCsFXS-iPSC-derived neurons# of CGG repeats in FXS-iPSCs may not equal the # of CGG repeats in fibroblasts of origin.Determined the methylation status of *FMR1* and FMRP expression.FXS-iPSC-derived neurons exhibited underdeveloped neurites.Doers et al. (2014) [48] ***Generation of human iPSC-derived cellular models of FXS***Characterization of FXS-iPSCs***Characterization of neurodevelopmental and electrophysiological abnormalities of iPSC-derived cell models of FXS***FXS-iPSCsFXS-iPSC-derived neurons*FMR1* repeat expansion and hypermethylation of the promoter is retained following the reprogramming of FXS patient fibroblasts to iPSCs.Axonal growth cones of FXS-iPSC-derived neurons have reduced motility and rates of extension compared to controls.Esanov et al. (2016) [49]***Generation of human iPSC-derived cellular models of FXS***Characterization of FXS-iPSCsPrimary neurons from postmortem FXS patient brain tissuePrimary fibroblasts from FXS patientsImmortalized lymphocytes from FXS patientsFXS-iPSCsFXS-iPSC-derived neuronsFXS-ESC-derived neurons*FMR1* promoter 5hmC enrichment was present in primary FXS neurons but is absent in primary FXS fibroblasts, lymphocytes, FXS-iPSCs, FXS-iPSC-derived neurons, and FXS-ES-derived neurons.De Esch et al. (2014) [50]***Generation of human iPSC-derived cellular models of FXS***Characterization of UFM iPSCsUFM iPSCsThe reprograming process from fibroblasts to iPSCs caused silencing of the fully mutated *FMR1* gene in UFM iPSCs.Brykczynska et al. (2016) [51]***Generation of human iPSC-derived cellular models of FXS***Characterization of UFM iPSCsUFM iPSCs*FMR1* in UFM iPSCs could be silenced when the gene contained CGG repeats above the threshold of around 400.Halevy et al. (2015) [52]***Gene expression and translation in the 2D hiPSC-derived cell models of FXS***FXS-iPSC-derived neuronsRE-1-silencing transcription factor (*REST*), which suppresses neuronal differentiation and axon guidance genes, was upregulated, and hsa-mir-382, which represses *REST*, was downregulated in FXS-iPSC-derived neurons.Overexpression of hsa-mir-382 significantly upregulated *REST*-mediated genes in FXS-iPSC-derived neurons.Utami et al. (2020) [53]***Gene expression and translation in the 2D hiPSC-derived cell models of FXS******Characterization of neurodevelopmental and electrophysiological abnormalities of iPSC-derived cell models of FXS***FXS-iPSC-derived NPCs & neuronsIsogenic *FMR1*-KO hESCs*FMR1*-KO hESC-derived NPCs & neuronsRNA-seq revealed that genes related to kinase activity, amino acid transport, and RNA methylation were upregulated in FXS-iPSC-derived neurons. Genes related to axon guidance, neuron differentiation, transsynaptic signaling, and messenger RNA splicing were downregulated in FXS-iPSC-derived neurons.Significantly smaller neural rosettes were formed by FXS-iPSCs compared to controls.Proliferation was increased in FXS-iPSC-derived NPCs compared to controls.Neurite outgrowth was decreased in FXS-iPSC-derived neurons compared to control neurons.Lu et al. (2016) [54]***Gene expression and translation in the 2D hiPSC-derived cell models of FXS***FXS-iPSCsFXS-iPSC-derived neuronsRNA-seq revealed upregulated genes related to neuronal differentiation and neural development and downregulated genes encoding potassium channels in FXS-iPSC-derived neurons.Boland et al. (2017) [55]***Gene expression and translation in the 2D hiPSC-derived cell models of FXS******Characterization of neurodevelopmental and electrophysiological abnormalities of iPSC-derived cell models of FXS***FXS-iPSC-derived NPCs & neurons*REST* target genes were dysregulated in FXS-iPSC-derived NPCs and neurons.Many DEGs associated with developmental signaling and cell migration were identified in FXS-iPSC-derived neural cells.Immature neurons derived from FXS-iPSCs exhibited increased neurite lengths compared to control neurons.Sunamura et al. (2018) [56]***Gene expression and translation in the 2D hiPSC-derived cell models of FXS****FMR1*-KO hiPSC-derived NPCs*FMR1*-null NPCs exhibited:Elevated expression of glial fibrillary acidic protein (GFAP), which was corrected by reintroduction of FMRP via a lentiviral vector.Reduced spontaneous calcium burstsBoth of the above abnormalities were corrected by treatment with protein kinase inhibitor LX7101.Raj et al. (2021) [21]***Gene expression and translation in the 2D hiPSC-derived cell models of FXS******Modeling FXS with human brain organoids***FXS-iPSC-derived NPCsIsogenic *FMR1*-KO iPSC-derived NPCsFXS-iPSC-derived cortical organoids at differentiation day 28The authors developed a flow-cytometry-based high-throughput single-cell assay to measure translation and proliferation markers specific to neural cell type.FXS NPCs expressed higher levels of markers of proliferation than those of neurogenesis.An inhibitor of a catalytic subunit of PI3K ameliorated the protein synthesis defects in FXS NPCs.FXS organoids showed increased NPC proliferation.Transcriptome analysis of FXS organoids identified significantly upregulated genes to be related to proliferation and significantly downregulated genes to be related to neuronal fate specification, migration, differentiation, and maturation.Kurosaki et al. (2021) [57]***Gene expression and translation in the 2D hiPSC-derived cell models of FXS***FXS-iPSCsFXS-iPSC-derived neuronsFMRP represses nonsense-mediated mRNA decay (NMD).Small molecules that inhibit NMD restored the aberrant expression of neurodifferentiation markers and increased the neurite growth in FXS-iPSC-derived neurons.Li et al. (2020) [23]***Studies of FMRP function and targets***Dorsal and ventral forebrain NPCS & neurons derived from FMR1-FLAG tagged and *FMR1*-KO hPSCs (iPSCs and ESCs)CLIP-seq identified FMRP targets common to all four types of cells as well as cell type-specific targets.An integrative analysis of CLIP-seq and transcriptomic data revealed *FMR1*-regulated pathways essential in human neurodevelopment.FMRP bound preferentially to longer RNA targets and to coding regions of mRNAs.Goering et al. (2020) [58]***Studies of FMRP function and targets***FXS-iPSC-derived neuronsFMRP-KO mouse CAD neuronal cellsFMRP regulates the localization of certain gene transcripts to neurites within neurons, and these FMRP-localization target genes were enriched with G-quadruplex structures in their 3′ UTRs.Localization targets of FMRP differed from translation targets of FMRP.Niedringhaus et al. (2015) [90]***Characterization of neurodevelopmental and electrophysiological abnormalities of iPSC-derived cell models of FXS***FXS-iPSC-derived neuronal cultures using a micro-raft arrayFXS-iPSC-derived neurons had a significant decrease in synaptic vesicle recycling and an increase in unloading of synaptic vesicles compared to control neurons.Zhang et al. (2018) [60]***Characterization of neurodevelopmental and electrophysiological abnormalities of iPSC-derived cell models of FXS***FXS-iPSC-derived neuronsFXS-ESC-derived neuronsBaseline amplitude and frequency of miniature excitatory postsynaptic current (mEPSC) and inhibitory equivalent (mIPSC) did not differ between FXS neurons and controls.FXS NPCs exhibited impaired retinoic-acid-mediated regulation of synaptic strengths, which was rescued by the recovery of *FMR1* expression by excision of the expanded CGG repeats using CRISPR/Cas9.Das Sharma et al. (2020) [61]***Characterization of neurodevelopmental and electrophysiological abnormalities of iPSC-derived cell models of FXS***FXS-iPSC-derived neuronsFXS-iPSC-derived neurons fired shorter and more frequent spontaneous action potentials than *FMR1*-expressing controls.Voltage-gated sodium (Na^+^) channel activator treatment increased the duration and reduced the frequency of action potentials in FXS neurons, normalizing the firing patterns to resemble those in controls.In contrast, treatment of control lines with a persistent Na^+^ current (Na_p_) blocker and a calcium (Ca^2+^)-activated potassium (K_Ca_) channel blocker both altered the firing patterns to resemble those in FXS neurons.Danesi et al. (2018) [63]***Characterization of neurodevelopmental and electrophysiological abnormalities of iPSC-derived cell models of FXS***FXS-iPSC-derived NPCs*Fmr1*-KO and dMT mouse NPCsIntracellular calcium release signaling in response to membrane depolarization and the expression of L-type Ca_v_ channels were increased in both human and mouse NPCs lacking *FMR1* expression at an early stage of differentiation.Achuta et al. (2017) [62]***Characterization of neurodevelopmental and electrophysiological abnormalities of iPSC-derived cell models of FXS***FXS-iPSC-derived NPCs*Fmr1*-KO mouse NPCsFXS-hiPSC-derived NPCs and *Fmr1*-KO mouse NPCs both exhibited larger intracellular calcium release amplitudes in response to the group I mGluR (mGluR1 and mGluR5) agonist (*S*)-3,5-dihydroxyphenylglycine (DHPG) compared to controls.MPEP affects neural development in a species-dependent and cell-type-dependent manner.Achuta et al. (2018) [59]***Characterization of neurodevelopmental and electrophysiological abnormalities of iPSC-derived cell models of FXS***Maturation of excitatory transmission in FXSFXS-iPSC-derived NPCs*Fmr1*-KO mouse NPCsFXS-iPSC-derived NPCs and *Fmr1*-KO mouse NPCs express a higher level of Ca^2+^-permeable AMPA receptors (CP-AMPARs) lacking the GluA2 subunit than controls.Naspm, a Glu2-lacking CP-AMPAR inhibitor, reduced the lengths of neurites in both FXS and control neurons so that their lengths were not significantly different from each other.Brighi et al. (2021) [24]***Characterization of neurodevelopmental and electrophysiological abnormalities of iPSC-derived cell models of FXS***Maturation of excitatory and inhibitory transmission in FXS***Modeling FXS with human brain organoids***FMRP-KO-hiPSC-derived neuronsFMRP-KO-hiPSC-derived organoids*FMR1*-null NPCs had elevated expression of glial fibrillary acidic protein (GFAP).*FMR1*-null NPCs exhibit increased spontaneous electrophysical network activity and depolarization in response to GABA.FMRP-KO organoids were bigger in size and had an increased number of glial cells, presumably astrocytes.Bar-Nur et al. (2012) [64]***FMR1 reactivation—Pharmacological rescue***FXS-iPSCsFXS-iPSC-derived neurons5-azaC treatmentReactivated *FMR1* expression in both FXS-iPSCs and FXS-iPSC-derived neurons.Reduced the methylation of the *FMR1* promoter region in FXS-iPSCs and neurons in a concentration-dependent manner.Led the histone H3 acetylation and H3K4 methylation levels to be comparable to those in controls.Kaufmann et al. (2015) [65]***FMR1 reactivation—Pharmacological rescue***FXS-iPSC-derived NPCsA high-content imaging assay was developed and used to screen compounds that could reactivate the expression of *FMR1*. Several compounds that led to a weak expression of FMRP were identified.Kumari et al. (2015) [67]***FMR1 reactivation—Pharmacological rescue***FXS-iPSC-derived neural stem cellsA time-resolved fluorescence resonance energy transfer (TR-FRET) assay was developed and used to screen compounds that could reactivate the expression of *FMR1*. Six compounds that could partially reactivate *FMR1* were identified.Li et al. (2017) [68]***FMR1 reactivation—Pharmacological rescue***FXS-iPSCsHuman iPSC reporter line with a CRISPR/Cas9-based knock-in of a Nano luciferase (Nluc) gene (*FMR1*-Nluc reporter line) to detect *FMR1* expression was created.Vershkov et al. (2019) [69]***FMR1 reactivation—Pharmacological rescue***FXS-iPSCsFXS-iPSC-derived NPCsFXS-iPSC-derived NPC transplants in mice*FMR1* reactivation in FXS-iPSCs and FXS-iPSC-derived NPCs by 5-azadC was enhanced by the addition of 3-deazaneplanocin A (DZNep).*FMR1* reactivation in response to 5-azadC was verified in vivo in FXS-iPSCs injected into immunocompromised mice and FXS-iPSC-derived NPCs transplanted into mouse brains.Kumari et al. (2020) [66]***FMR1 reactivation—Pharmacological rescue***FXS-iPSC-derived neural stem cells & neuronsChaetocin, a fungal toxin that inhibits mammalian histone methyl-transferases, had a synergistic effect with 5-azadC in reactivating *FMR1* in neural stem cells and neurons derived from FXS-iPSCs.Chaetocin, DZNep, and BIX01294 delayed the re-silencing of 5-azadC-activated *FMR1* expression.Park et al. (2015) [70]***FMR1 reactivation—Gene editing***FXS-iPSCsFXS-iPSC-derived neuronsExpanded CGG repeats in *FMR1* were excised by CRISPR/Cas9 in FXS-iPSCs, which led to the near-complete demethylation and expression of FMR1, and FMRP expression in FXS-iPSCs and CGG-repeat-edited-FXS-iPSC-derived neurons.Xie et al. (2016) [71]***FMR1 reactivation—Gene editing***FXS-iPSCsSomatic cell hybrids containing a human FXS chromosomeExpanded CGG repeats in *FMR1* were excised by CRISPR/Cas9 in FXS-iPSCs, which led to the demethylation and expression of *FMR1*, and FMRP expression in FXS-iPSCs and FXS somatic hybrid cells.Liu et al. (2018) [72]***FMR1 reactivation—Gene editing***FXS-iPSCsFXS-iPSC-derived neuronsFXS-iPSC-derived NPC transplants in miceExpanded CGG repeats were demethylated using CRISPR/dCas9-Tet1 in FXS-iPSCs.Targeted methylation editing led to an active chromatin state of *FMR1* promoter, normalization of electrophysiological abnormalities in neurons derived from methylation-edited FXS-iPSCs.*FMR1* reactivation was sustained in methylation-edited neuronal engrafts in mice.Graef et al. (2020) [73]***FMR1 reactivation—Gene editing***FXS-iPSCsIsogenic *FMR1*-KO iPSCsFXS-iPSC-derived neurons5% of normal FMRP expression was enough to rescue the elevated spontaneous activity in an FXS mosaic neuronal culture. Moreover, a neuronal culture in which greater than 20% of cells express FMRP had a normal electrophysiological phenotype.Kang et al. (2021) [22]***Modeling FXS with human brain organoids***FXS-iPSC-derived forebrain organoidsFXS organoids at differentiation day 56 (D56) exhibited reduced NPC proliferation, premature neural differentiation, altered cortical layer formation, and disrupted differentiation of GABAergic interneurons.FXS organoids at D56 showed accelerated synapse formation and hyperexcitabilityFXS organoids had altered gene expression profiles and aberrant cell-type-specific developmental trajectory.PI3K inhibitors but not mGluR5 antagonists rescued NPC proliferation defects and synaptic formation deficits in FXS organoids.A large number of human-specific FMRP targets were identified via eCLIP-seq, including chromodomain helicase DNA-binding protein 2 (CHD2)FXS = fragile X syndrome, iPSCs = induced pluripotent stem cells, FXS-iPSCs = iPSCs derived from primary cells from individuals with FXS, ESC = embryonic stem cell, NPC = neural progenitor cell, UFM = unmethylated full mutation, MPEP = 2-methyl-6-(phenylethynyl)pyridine, 5-azaC = 5-azacytidine, 5-azadC = 5-aza-2′-deoxycytidine or decitabine, CRISPR = clustered regularly interspaced short palindromic repeats, eCLIP-seq = enhanced crosslinking and immunoprecipitation followed by high-throughput sequencing.

### 2.4. Characterization of Neurodevelopmental and Electrophysiological Abnormalities of iPSC-Derived Cell Models of FXS

Many studies have focused on the neurodevelopmental features of FXS-iPSCs and iPSC-derived neural cells compared to those of controls. Several groups have found that FXS-iPSC-derived neurons have altered neurite development compared to controls (Figure 1D) [47,48,53,55,59]. Three studies found that FXS neurons have underdeveloped neurites with shorter and fewer neural processes, indicating impairment in both initiating the development of neurites and extending them [47,48,53]. Doers et al. (2014) also reported that axonal growth cones of FXS-iPSC-derived neurons have reduced motility and rates of extension [48]. Another study, through transcriptomic and proteomic analyses, found that genes involved in regulating axonal growth were downregulated in FXS-iPSCs compared to controls [53], which could further explain the undergrowth of neurites. This study also showed that FXS-iPSCs formed significantly smaller neural rosettes than control iPSCs did [53]. However, one study found that immature neurons derived from FXS-iPSCs, in fact, exhibit increased neurite lengths compared to neurons derived from *FMR1*-expressing iPSCs [55]. The authors state that the diverging results are likely due to differing maturity of the neurons used in the studies. One gene possibly playing a role is *SLITRK4*, which restrains the growth of neurites, found to be downregulated in NPCs but not in mature neurons [55].

Another group, using a micro-raft array to generate large-scale neuronal cultures, showed that FXS-iPSC-derived neurons have a significant decrease in synaptic vesicle recycling and an increase in unloading of synaptic vesicles compared to control iPSC-derived neurons [90]. These findings were also seen in primary neurons from *Fmr1*-KO mice compared to those from wildtype mice [90].

Several studies have characterized the electrophysiological properties of FXS-iPSCs and iPSC-derived neural cells (Figure 1E) [24,59,60,61,62,63]. One study found altered homeostatic synaptic plasticity in FXS-iPSC-derived neurons compared to controls [60]. Although FXS neurons did not differ from controls in their baseline amplitude and frequency of miniature excitatory postsynaptic current (mEPSC), nor the inhibitory equivalent (mIPSC), FXS NPCs exhibited impaired retinoic-acid-mediated regulation of synaptic strengths. The increase in mEPSC amplitudes and decrease in mIPSC amplitudes driven by retinoic acid treatment seen in control NPCs were absent in FXS NPCs. Moreover, recovery of *FMR1* expression by excision of the expanded CGG repeats using CRISPR/Cas9 normalized the retinoic-acid-mediated synaptic signaling [60].

Whereas the intrinsic properties of mEPSCs or mIPSCs did not appear to differ between FXS and control groups [60,61], FXS-iPSC-derived neurons were found to fire shorter and more frequent spontaneous action potentials than *FMR1*-expressing controls [61]. The authors then tested the effects of pharmacological treatment on action potential firing in both FXS neurons and controls. While voltage-gated sodium (Na^+^) channel activator treatment had no significant effect on control neurons, it increased the duration and reduced the frequency of action potentials in FXS neurons, normalizing the firing patterns to resemble those in controls. Furthermore, the treatment of control lines with a persistent Na^+^ current (Na_p_) blocker and a calcium (Ca^2+^)-activated potassium (K_Ca_) channel blocker decreased the duration and increased the frequency of action potential bursts, respectively, both altering the firing patterns to resemble those in FXS neurons [61]. The authors highlight that the study finding of the FXS group displaying a lower conductance of K_Ca_ current mediated by the binding of FMRP to the β4 subunit of the K_Ca_ channel is consistent with findings in *Fmr1*-KO mice [91,92]. However, the finding that there is reduced Na_p_ conductance in FXS-iPSC-derived neurons contrasts with findings from mouse models of FXS with increased Na_p_ conductance [93,94], highlighting possible species-specific differences in pathophysiology.

One group that focused on voltage-gated calcium (Ca_v_) channels in FXS-iPSC-derived NPCs and *Fmr1* KO mouse NPCs found that the intracellular calcium release signaling in response to membrane depolarization was increased in both human and mouse NPCs lacking *FMR1* expression at an early stage of differentiation (day 1) [63]. Moreover, in both humans and mice, the expression of L-type Ca_v_ channels was increased in FXS NPCs, indicating the role of this channel type in the enhanced intracellular calcium response [63]. The same group also found that both human FXS NPCs and *Fmr1*-KO mouse NPCs exhibited larger intracellular calcium release amplitudes in response to the group I mGluR (mGluR1 and mGluR5) agonist (*S*)-3,5-dihydroxyphenylglycine (DHPG) compared to controls and WTs, respectively [62]. Based on previous research that metabotropic glutamate receptor 5 (mGluR5) signaling is altered in FXS [95,96], the authors investigated the effects of a selective mGluR5 antagonist 2-methyl-6-(phenylethynyl)pyridine (MPEP) on neuronal differentiation in FXS hiPSC-derived NPCs and *Fmr1*-KO mouse NPCs [62]. In both human and mouse FXS NPCs, the proportion of cells responsive to DHPG and ionotropic glutamate receptors (iGluRs) was increased compared to controls. Interestingly, however, whereas in human NPCs, MPEP increased the differentiation of NPCs to cells responsive to DHPG and those responsive to both DHPG and iGluRs, MPEP decreased the differentiation of *Fmr1*-KO mouse NPCs to cells responsive to iGluRs, with no effect on differentiation of DHPG-responsive cells. Furthermore, human FXS NPCs had a larger subpopulation of cells responsive to kainate and NMDA than control NPCs did, but this difference between groups was absent in mouse NPCs. MPEP normalized the increase in NMDA-responsive cells in the FXS group but reduced the kainate-responsive cells in both FXS and control groups. These findings imply that MPEP affects neural development in a species-dependent and cell-type-dependent manner [62] and that glutamatergic signaling pathways may be regulated differently.

Two studies have revealed impairments in maturation of excitatory [24,59] and inhibitory [24] transmission in FXS using iPSC-derived models. One study (Achuta et al., 2018) showed that FXS-iPSC-derived NPCs express a higher level of Ca^2+^-permeable AMPA receptors (CP-AMPARs) lacking the GluA2 subunit than the controls, representing an aberrant maturation of this subtype of glutamatergic transmission system in FXS [59]. The authors also confirmed this finding in *Fmr1*-KO mouse NPCs, showing that the process is conserved across species. GluA2-lacking CP-AMPARs are typically found in early developmental stages of neural progenitor differentiation [97,98,99,100]. A larger proportion of FXS NPCs responded to Naspm trihydrochloride (Naspm; a Glu2-lacking CP-AMPAR inhibitor) treatment, also supporting that there are elevated levels of this AMPA receptor type in the FXS group. Based on previous findings that AMPARs can modulate neurite growth [98,99], the authors tested the effect of Naspm on neurite length in iPSC-derived neurons. Naspm reduced the lengths of neurites in both FXS and control neurons so that their lengths were not significantly different from each other, demonstrating that CP-AMPARs have a role in regulating the growth of neural processes [59]. Interestingly, *GRIA2*, which encodes GluA2, is a target of REST, whose interaction with FMRP is mediated by the miRNA machinery [52]. The authors explain that the lack of Glu2A in FXS NPCs could be due to increased expression of a microRNA miR-181a repressing Glu2A as a consequence of the absence of FMRP [59].

In line with findings by Achuta et al. (2018) of abnormal glutamatergic transmission in FXS [59], Brighi et al. (2021) found that a local application of glutamate triggered a significantly higher increase in intracellular calcium levels in neurons derived from *FMR1*-knockout human iPSC than in FMRP-expressing neurons [24]. The group also revealed aberrant development of GABA, the main inhibitory neural transmitter, signaling in FMRP-null neurons. Whereas focal GABA application elicited depolarization in about 30% of FMRP-expressing neurons, about 75% of FMRP-null neurons depolarized in response, which indicates that there is a delay in the transition of GABA transmission from excitatory to inhibitory. The GABA and chloride equilibrium reversal has been found to be delayed in *Fmr1*-KO mice [101] as well.

### 2.5. Reactivation of FMR1 Gene in FXS

Given that the silencing of *FMR1* leads to significant defects in neurodevelopment, many studies have aimed to reactivate *FMR1*. There are two main approaches, pharmacological agents and gene editing (Figure 1F).

#### 2.5.1. FMR1 Reactivation—Pharmacological Rescue

The most studied pharmacological compounds are the DNA methyltransferase (DNMT) inhibitor 5-azacytidine (5-azaC, azacytidine) and its deoxy derivative 5-aza-2′-deoxycytidine (5-azadC, decitabine) [10,64,65,67,68,69]. In an early study using 5-azaC in FXS-iPSCs and FXS-iPSC-derived neurons, 5-azaC treatment reactivated *FMR1* expression to about 15% to 45% of wildtypes (WTs) in FXS-iPSCs [64]. *FMR1* expression was restored by 5-azaC in FXS-iPSC-derived neurons as well, but to a lesser degree. The authors describe that 5-azaC can repress DNMT3a and DNMT3b, which are expressed primarily in pluripotent stem cells, possibly explaining the difference in efficacy of 5-azaC in the two cell types. Furthermore, *FMR1* reactivation by 5-azaC was retained for at least 7 days after treatment was stopped (not tested past 7 days). 5-azaC treatment substantially reduced the methylation of the *FMR1* promoter region in FXS-iPSCs and neurons in a concentration-dependent manner, with the effect being more profound in iPSCs than in neurons. Moreover, following 5-azaC treatment, histone H3 acetylation and H3K4 methylation levels were found to be comparable to those in WT [64].

Two groups independently developed high-throughput drug screening assays to measure *FMR1* reactivation in FXS-iPSC-derived neural cells, using high-content imaging [65] or time-resolved fluorescence resonance energy transfer (TR-FRET) [67]. Both studies were to provide proof of principle that the assays are feasible methods for screening possible treatment agents. Although the studies identified a few promising compounds that reactivated *FMR1* expression, none were as effective as 5-azaC or 5-azadC without being cytotoxic [65,67].

A few studies have demonstrated methods to further advance high-throughput drug screening. One group developed micro-raft arrays that can be used to expedite screening assays by creating homogenous neuronal cultures at a larger scale. The approach was tested to culture FXS-iPSC-derived neurons and comparing their synaptic features to control iPSC-derived neurons (see Section 2.4 for a summary of findings). Another group created a human iPSC reporter line through a CRISPR/Cas9-based knock-in of a Nano luciferase (Nluc) gene into the human *FMR1* gene (*FMR1*-Nluc reporter line), which can be used to detect *FMR1* gene expression with high sensitivity [68].

A small-molecule screening study revealed that *FMR1* reactivation in FXS-iPSCs and FXS-iPSC-derived NPCs by 5-azadC was enhanced if used in combination with 3-deazaneplanocin A (DZNep), an inhibitor of both S-adenosyl-homocysteine (SAH) hydrolase and histone methylation [69]. *FMR1* reactivation in response to 5-azadC was also verified in vivo in (1) FXS-iPSCs that were injected into immunocompromised mice and (2) FXS-iPSC-derived NPCs transplanted into mouse brains. Moreover, the reactivation was maintained even after 30 days of treatment withdrawal [69]. Another study of small-molecule inhibitors of histone methyl-transferases (HMTs) showed that chaetocin, a fungal toxin that inhibits four main mammalian HMTs (G9a, GLP, SUB39H1, and SETDB1), had a synergistic effect with 5-azadC in reactivating *FMR1* in neural stem cells (NSCs) and neurons derived from FXS-iPSCs [66]. These studies highlight the interplay of DNA methylation and histone modifications contributing to *FMR1* gene silencing and consequent FXS pathogenesis.

#### 2.5.2. FMR1 Reactivation—Gene Editing 

Gene editing approaches to reactivate *FMR1* have led to a better understanding of the relationship between CGG repeat expansions and methylation of *FMR1*, as well as facilitating the development of methods for a more stable restoration of *FMR1* expression than by pharmacological agents.

Two groups have shown that reactivation of *FMR1* in FXS-iPSCs [70,71] and FXS-iPSC-derived neurons [70] can be achieved by excising out the expanded CGG repeats using CRISPR/Cas9, confirmed by demethylation of the *FMR1* promoter, transition to an active chromatin state, and FMRP expression [70,71]. Park et al. (2015) used a double-strand break (DSB) followed by nonhomologous end joining of sequences flanking the CGG repeats [70], whereas Xie et al. (2016) used two DSBs to flank CGG repeats [71]. The former group selected wildtype clones with about 90bp of CGG repeat deletion from their parental lines as the control group, and FXS iPSC clones with similar-sized PCR products to those of the controls as the comparison group. This process resulted in the selection of 2–3 clones out of approximately 100 colonies. On the other hand, the latter group identified five clones with CGG repeat deletions by CRISPR, and only one among the five displayed both *FMR1* reactivation and FMRP expression. Although the efficiencies cannot be directly compared between the two studies since the methods and techniques differ, the low overall efficiency of the approach using CRISPR editing to remove CGG repeat expansions warrants further research.

Another group used a DNA methylation editing method by CRISPR/dCas9-Tet1 system to remove the hypermethylation in CGG repeat expansions of the *FMR1* locus in FXS-iPSCs and postmitotic neurons derived from FXS-iPSCs [72]. Following targeted demethylation, the *FMR1* promoter region transitioned from a heterochromatin state to an active chromatin state, and *FMR1* expression was restored in both cell types, although to a lesser extent in postmitotic neurons compared to iPSCs. Furthermore, neurons that were derived from the methylation-edited FXS-iPSCs exhibited normalized electrophysiological phenotypes as examined by a multi-electrode array assay. Persistent expression of *FMR1* was detected in vivo in methylation-edited FXS-iPSC-derived neurons transplanted in mouse brains [72].

A recent study measured the threshold of *FMR1* reactivation to normalize FXS disease phenotype using a combination of techniques, including CRISPR and antisense oligonucleotides [73]. Results demonstrated that 5% of normal FMRP expression was enough to rescue the elevated spontaneous activity in a mosaic culture of excitatory neurons derived from FXS-iPSCs and *FMR1*-expressing neurons. Moreover, a neuronal culture in which more than 20% of cells express FMRP had a normal electrophysiological phenotype. However, the reactivation of FMRP was studied in one isogenic line and, thus, verification of results using several different patient cell lines is necessary [73].

As outlined above, both pharmacological and genetic-editing-based reactivation of *FMR1* have led to promising results. Partial reactivation of *FMR1* may be enough to rescue critical neurodevelopmental abnormalities seen in FXS. Future studies using a standardized protocol for cell cultures and *FMR1*/FMRP detection assays comparing the efficacies of different genetic editing techniques and their combinations will elucidate the contribution of the different epigenetic mechanisms of FXS pathogenesis. Small molecule screening studies can reveal critical targets interacting with FMRP, which can, in turn, drive genetic editing studies for *FMR1* reactivation. Gene editing studies can conversely guide the development of new pharmacological agents specific for FXS.

## 3. Three-Dimensional (3D) hiPSC-Derived Models of FXS 

While 2D models of FXS yielded many remarkable findings of the disease mechanisms, they are somewhat limited as having only one or a few (if co-cultured) cell types for investigation. This does not accurately replicate what is seen in the human brain, where different types of neurons and glial cells co-exist and co-ordinate with one other, forming functional areas that control biological activities in the body [102,103,104]. Therefore, a more sophisticated model system is needed to better recapitulate human neurodevelopment. The development of a recent advanced technique, brain organoids, provides such an opportunity and allows for more accurate modeling of FXS in vitro.

### 3.1. Brain Organoids—In Vitro System Recapitulating Human Neurodevelopment

Brain organoids are human pluripotent stem cells (hPSC)-derived 3D suspension cultures that recapitulate many features of the developing brain, such as ventricular zone formation, neurogenesis, neuronal migration, transcriptomic and epigenomic profiles, and gliogenesis under long-term culturing [105,106,107,108,109,110,111,112,113]. Brain organoids, therefore, provide a unique platform for investigating normal and abnormal neurodevelopment. The two main categories of brain organoids are (1) cerebral organoids (also known as whole brain organoids), generated via unguided methods that provide no external cues and allow for spontaneous differentiation and self-organization within the organoids [106,112,114,115], and (2) brain-region-specific organoids, generated via protocols guiding the addition of patterning factors (i.e., growth factors and small molecules) to promote the differentiation to a specialized brain region [105,107,110,116,117,118]. Examples of currently available brain-region-specific organoids include forebrain, midbrain, hippocampus, thalamus, hypothalamus, cerebellum, choroid plexus, and ganglionic eminence organoids [105,110,118,119,120,121,122,123,124]. The two types of brain organoids harbor distinct features and advantages/disadvantages. Cerebral organoids, being generated by the spontaneous differentiation capacity of hPSCs, contain a variety of cell lineages, ranging from dorsal forebrain, ventral forebrain, midbrain, hindbrain, and hippocampus, to choroid plexus, retina, astrocytes, and oligodendrocyte [106,112,125,126,127,128,129], whereas brain-region-specific organoids contain only the cell types that are present in the region of interest. Cerebral organoids are, thus, more advantageous in cell lineage diversity, providing opportunities for studying region–region interactions during neurodevelopment. However, the stochastic nature of spontaneous differentiation of hPSCs also brings in additional randomness and heterogeneity to cerebral organoids—i.e., unpredictable and inconsistent proportion and organization of cell lineages within each organoid, making systematic and quantitative analysis difficult and less reliable [130]. From this perspective, brain-region-specific organoids are more advantageous as they contain relatively consistent cell populations with less variability among individuals, improving the reproducibility and reliability of the experiments.

### 3.2. Modeling FXS with Human Brain Organoids

The first study utilizing human brain organoids as a disease model of FXS made use of patient iPSC-derived cortical organoids [21]. Raj et al. (2021) found that FXS cortical organoids at differentiation day 28 exhibited a higher percentage of Ki67^+^SOX2^+^ proliferative cells compared to the controls, supporting their findings in 2D cellular models that loss of FMRP led to abnormal proliferation and cell cycle alterations in FXS [21]. Moreover, transcriptomic analysis also revealed a large number of differentially expressed genes (DEGs) between control and FXS cortical organoids. Gene ontology (GO) analysis identified significantly upregulated genes in FXS to be related to proliferation, while significantly downregulated genes in FXS were related to neuronal fate specification, migration, differentiation, and maturation (Figure 2). These results together suggested altered cell-fate decisions in FXS to favor proliferation over differentiation during early neurodevelopment in humans.

FXS cerebral organoids derived from a CRISPR/Cas9-mediated FMRP-KO hiPSC line were investigated in another study [24]. Brighi et al. (2021) found that FMRP-KO organoids not only were bigger in size compared to control organoids, but also had an increased number of glial cells, presumably astrocytes, suggesting an important role of FMRP in regulating gliogenesis and the balance of neural and glial development, as shown with 2D FXS iPSC-derived cellular models (Figure 2) [24,56]. However, the molecular and cellular mechanisms underlying the observed phenotypes were not elucidated.

These questions were investigated in depth in another study published recently [22]. Kang et al. (2021) utilized FXS patient iPSC-derived forebrain organoids for a comprehensive and thorough investigation on how FMRP deficiency affects gene expression, cellular composition, and propagation, as well as neuronal function during human cortical development [22]. FXS forebrain organoids derived from patient iPSC lines were examined at differentiation day 56 (D56), a critical time period that strongly correlates with the mid-fetal period in humans when cortical neurogenesis and FMRP depletion are in play [110,131,132]. By immunostaining with various markers delineating major aspects of cortical development, the authors revealed that the loss of FMRP impaired neurogenesis and cortical development in many ways, including reduced neural progenitor cell (NPC) proliferation, premature neural differentiation, altered cortical layer formation, disrupted differentiation of GABAergic interneurons, as well as accelerated synapse formation that was consistent with the hyperexcitability found by electrophysiological analyses. Notably, as opposed to what was found in Raj et al. (2021), the reduction in NPC proliferation in these D56 FXS organoids could be suggesting a developmental-stage-specific alteration in NPC proliferation rate, which will be an interesting topic to investigate in future studies. At the molecular level, bulk and single-cell RNA-seq revealed pervasive alterations in gene expression profiles and cell-type-specific developmental trajectories in FXS forebrain organoids, which, in combination with other mutant phenotypes, led to the investigation of potential pharmacological approaches that could rescue the phenotypes observed in these FXS organoids. Previous studies on animal models suggested that both genetic and pharmacological inhibitions of mGluR5 receptor were able to rescue both behavioral and synaptic abnormalities caused by the excessive mGluR signaling in FXS [133,134,135]. Additionally, elevated PI3K signaling has been reported in previous FXS models, in which genetic reduction or pharmacological inhibition of PI3K were able to ameliorate disease phenotypes [21,75,81,82]. Therefore, the authors tested the effects of mGluR5 or PI3K inhibition in FXS forebrain organoids. They found that, while mGluR5 antagonists did not rescue mutant phenotypes in these FXS forebrain organoids, PI3K inhibitors rescued NPC proliferation defects and synaptic formation deficits, suggesting potential human-specific disease mechanisms that had not been seen in previous animal models and highlighting a potential therapeutic target for FXS in humans. The authors went on to ask whether there are human-specific FMRP targets in the FXS forebrain organoids and were able to identify a large number of human-specific FMRP targets via eCLIP-seq, one of which is chromodomain helicase DNA-binding protein 2 (CHD2), a well-known risk gene associated with epilepsy, autism spectrum disorder (ASD), and intellectual disability (Figure 2) [136,137]. Together, these studies provide new insights into the human-specific molecular mechanisms of FXS and highlight the potential therapeutic targets for FXS and ASD in general.

## 4. Limitations and Future Directions

The aforementioned findings using hiPSC-derived cellular models of FXS highlight human-specific molecular mechanisms underlying FXS. However, there are some limitations of the systems. One major limitation of using hiPSC-based cellular models is the difficulty of controlling the many sources of variability. Factors that introduce variability include the diversity in cell lines, differences in reprograming and differentiation protocols, and instability introduced in reprograming [47,48,138]. Variability is greater in brain organoids than in 2D cellular models. The use of multiple cell lines for both control and FXS groups is important to reduce the possibility of making conclusions based on findings specific to selective clones. Consistency in culture and differentiation conditions is also crucial. Detailed documentation of protocols and enabling public access to methods will enhance reproducibility.

3D cellular models are more advantageous in cell type diversity, cellular architecture, and developmental trajectory. However, there are still limitations with the current systems, one of which is the imperfect representation of cellular composition and organization. For example, Kang et al. (2021) reported the percentage of GABAergic inhibitory neurons to be about 10% in their forebrain organoids, which is significantly lower compared to the actual human brain, making investigations of excitation/inhibition balance (E/I balance) and neural network activity difficult. One feasible approach would be to generate fusion organoids, also known as assembloids [139,140,141,142], that have dorsal forebrain and ventral forebrain organoids fused together. Assembloids allow for a closer examination of the interactions between different cell types and brain regions during neurodevelopment. A recent study utilizing cerebral cortex–ganglionic eminence (Cx + GE) assembloids as a disease model of Rett syndrome revealed defects in the balance of excitatory and inhibitory synapses, GE-dependent hypersynchronous neural network activity, and GE-dependent epileptiform changes in Rett syndrome fusion organoids, thus supporting the feasibility of using assembloids in the study of FXS [143].

An underexplored area of research using hiPSC-based models of FXS is investigating the impact of FMRP deficiency on glial development and function. Very few studies have explored hiPSC-derived glial cells in the study of FXS. One group recently published a method to generate astrocytes with forebrain patterning from hPSCs (hASTRO) [144]. Using this method, the group found that the expression of urokinase plasminogen activator (uPA), a protease that regulates the degradation of the extracellular matrix, was increased in FXS hASTRO compared to control hASTRO [145]. Interestingly, the increase in uPA level was correlated with a reduced intracellular calcium response to depolarization in hASTRO [145], which is in contrast to the enhanced intracellular calcium release seen in FXS NPCs [63]. The authors stated that uPA signaling could play a role in modulating synaptic excitation/inhibition balance found to be altered in FXS [146]. The neuronal–glial interaction in FXS needs to be further explored using hiPSC-derived cellular techniques to provide a better understanding of human-specific pathophysiological processes.

In 3D cellular models, only astrocytes and oligodendrocyte progenitor cells have been identified in typical cortical organoids after long-term culturing [109,110,139,147]. While oligodendrocytes function in myelination and metabolic support of neurons, microglia play important roles in brain infections and inflammation. Therefore, the absence of mature oligodendrocytes and microglia could affect neurodevelopment in cortical organoids, leading to the idea of incorporating these essential glial cells into brain organoids for better modeling of neurodevelopment. Several strategies have been tested, including utilizing certain differentiation inducers and promyelinating drugs to promote differentiation and maturation of oligodendrocytes in brain organoids [148,149], as well as co-culturing microglia with brain-region-specific organoids to promote the migration and incorporation of these cells into organoids [150,151,152,153]. However, none of these systems have been used in the study of FXS. Filling this gap would be an intriguing next step.

Finally, how FMRP deficiency affects late-stage neurodevelopment in humans is not well understood. Since most brain organoids only model the early and mid-fetal stage of neurodevelopment [110,111], less is known on late-stage neurodevelopment in FXS. Long-term culturing of brain organoids is a highly challenging task not only because it is time-consuming and labor-intensive, but also because the insufficient delivery of oxygen and nutrients to the inner core of brain organoids hamper the growth of cells and lead to the build-up of a necrotic core inside the organoid [130,154]. Solutions to this issue mainly focus on incorporating a vasculature system into the organoid. Some strategies include co-culturing of brain organoids with endothelial cells [155], induction of endothelial cell differentiation in cerebral organoids [156], overexpression of genes involved in vascular endothelial cell development [157], and transplantation of organoids into an in vivo system (immunodeficient mice) [155]. Other solutions, such as a sliced neocortical organoid system [158], have also been proposed to overcome the diffusion limit in brain organoids. Additionally, the combination of 2D and 3D cellular models is helpful for studying mature cell types in FXS. A study that longitudinally examines 2D and 3D models at multiple developmental stages will be ideal. As we are still at the beginning of understanding FXS in human neurodevelopment, these advanced techniques would serve as essential tools for the investigations along the road.

In summary, future studies of human FXS cellular models should aim to (1) improve reproducibility, (2) enhance representation of neural network structure and organization through the incorporation of glial cell types and use of assembloids, and (3) combine 2D and 3D models to provide a comprehensive examination of neurodevelopment from early to late stages (Figure 3).

## Figures and Tables

**Figure 2 cells-11-01725-f002:**
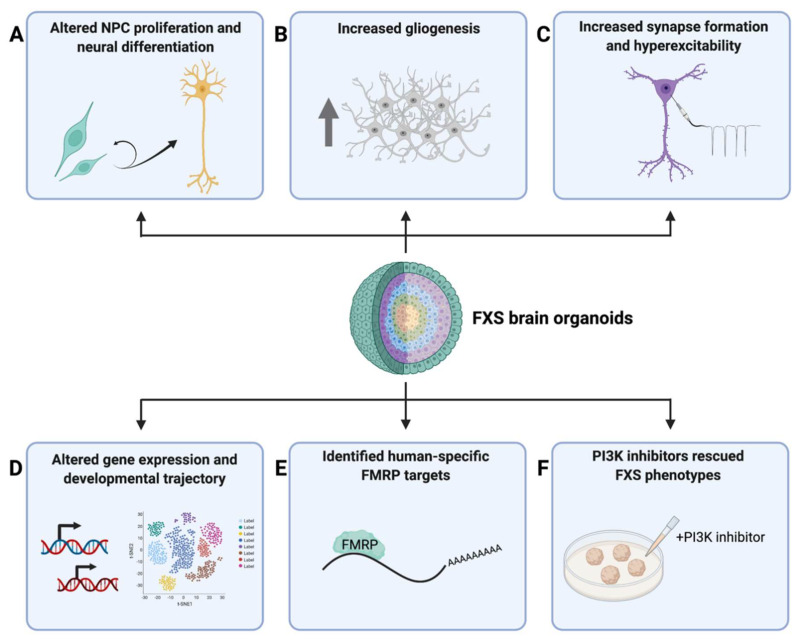
**Summary of major findings from 3D FXS−****iPSC−****derived models.** (**A**) FXS-iPSC-derived forebrain organoids exhibited altered NPC proliferation and neural differentiation [21,22]. In particular, increased NPC proliferation was observed at differentiation day 28 [21], while reduced NPC proliferation and accelerated neural differentiation were observed at day 56 [22], suggesting a developmental-stage-specific alteration in NPC proliferation rate. (**B**) Increased gliogenesis was observed in FMRP-KO cerebral organoids [24]. (**C**) FXS-iPSC-derived forebrain organoids showed increased synapse formation and hyperexcitability at differentiation day 56 [22]. (**D**) Transcriptome analysis revealed alterations in gene expression profile and cell-type-specific developmental trajectory in FXS-iPSC-derived forebrain organoids [21,22]. (**E**) Using human forebrain organoid models, a large number of human-specific FMRP targets were identified via eCLIP-seq [22]. (**F**) PI3K inhibitors were able to rescue FXS phenotypes in forebrain organoids [22].

**Figure 3 cells-11-01725-f003:**
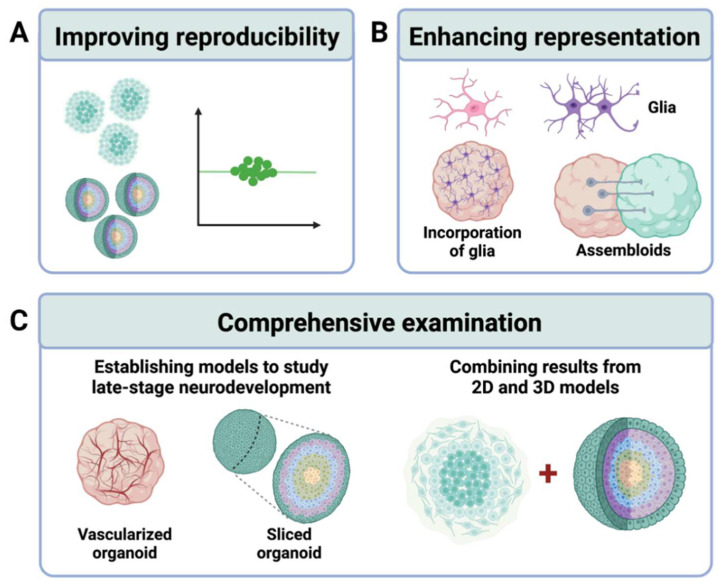
**Summary of future directions for FXS-iPSC research.** Future studies of human iPSC-based models of FXS should aim to (**A**) improve reproducibility, (**B**) enhance the representation of neural network structure and organization through the incorporation of various glial cell types and use of assembloids, and (**C**) provide a comprehensive examination of neurodevelopment from early to late stages by developing models that allow for prolonged culture, such as vascularized and sliced organoid models, as well as by corroborating results through a combination of 2D and 3D models.

## Data Availability

Not applicable.

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
