# Peer review of "Across Dimensions: Developing 2D and 3D Human iPSC-Based Models of Fragile X Syndrome"

_cells, 2022, doi:10.3390/cells11111725_

Round 1

Reviewer 1 Report

This review summarizes all studies that have used fragile X patient derived iPSCs for modeling all aspects of the disease.

The review covers the following topics:

  1. A brief introduction describing the molecular mechanism underlying fragile X syndrome and the consequences of loss of FMRP expression. It mentions all available models for FXS and emphasizes their limitations.
  2. Explains what are iPSCs, how are they established, and what are their apparent advantages.
  3. provides a comprehensive description of all relevant literature related to the study of FXS using iPSCs as a model system (summarized in Table 1)  including epigenetics, gene expression and cellular phenotypes following differentiation into neurons. In addition, it describes attempts for restoring gene expression by several  approaches.
  4. A summary of all recent work that relates to the use of brain organoids for improving disease modeling with FXS iPSCs.
  5. Concluding section that relates to the limitations of the current iPSCs-based models and future directions.

This review is clearly written. It systematically covers all the literature that relates to the use of iPSCs in the context of FXS. However, in my opinion it is not appropriate for publication in its current form for the following reasons: (1)    The authors disregard in their review the contribution of FXS hESCs to the study of FXS and their advantages (along with disadvantages) over patient derived iPSCs. This, to my opinion, reflects a lack of understanding regarding the fundamental differences between these two pluripotent stem cell types, especially when it comes to epigenetics regulated diseases. 

(2)  The review does not discuss the research findings in a critical or intelligent way. It is overloaded with information but often does not really examine the structure of the experiments or the data and its interpretation. One example out of a substantial number to illustrate: the authors describe both attempts for re-activating the FMR1 gene by deleting the CGGs with CRISPR/Cas9 in FXS iPSCs. However, they do not critically discuss the inconsistency between the reports and their meaning.  While in one study de-methylation was accompanied by a nearly 100bp deletion from the upstream flanking region (Park et al. 2015), in the other study (Xie et al. 2016) the excision of the CGGs resulted in de-methylation in only one out of the five FXS iPSC whith re-activated FMR1 gene. In addition, the authors did not point out that in Xie et al. a considerable number of successfully edited clones remained hypermethylated, implying that epigenetic resetting is not always efficient despite the repair of the DNA sequence.

(3)  The manuscript is written in a very subjective way and overemphasizes the advantage of iPSCs over other currently available animal model systems. The authors should “lower their tones” and emphasize the power of pluripotent stem cells as a complement model system.  

Reviewer 2 Report

I thoroughly enjoyed reading this comprehensive review.  It was well-structured and covered the topic in discussion in an engaging an interesting manner.  Whilst technical in its handling of the subject; it was also accessible enough for a broader audience.

I have no suggestions on how to improve it further.

Author Response

We appreciate the reviewer taking the time to critically review our manuscript and thank the reviewer for their positive assessment of our manuscript.

Reviewer 3 Report

This is an interesting, timely and thorough review of the literature involving Fragile X (FX) iPSCs. It explores the different 2D and 3D models and what we learned from them, the phenotypes (both molecular and cellular) that are associated with FX, the attempts to reactivate the expanded and methylated FMR1, and the limitations of the current systems.

The review, including the figures will be ueful for the scientific community.

One suggestion involves the Table. Table 1 is a very useful resource, but it is not clear how the items were ordered. If the authors choose not to list them chronologically, then some other logic should serve, which is currently not explained. Another option is to split the table into different “themes” (e.g. system-based, NPCs / neurons / organoids, isogenics, or based on the discoveries presented).

Round 2

Reviewer 1 Report

The authors significantly improved the manuscript although they are still missing important citations of previous work related to impaired neural function using mutant hESCs as a model system. For example, they did not relate to the work by Telias et al. 2015. Nor do they discuss or even mention the results of Gildin et al. Int. J. Mol. Sci. (2022) which show impaired action potentials and asynchronous network connectivity in mature fragile X induced neurons (FX-iNs). 

Author Response

Again, we appreciate the reviewer’s time taken to critically review our manuscript and thank the reviewer for their comments. We have added more citations of studies that have used hESC models to study FXS, including one suggested by the reviewer. Since the focus of our manuscript is iPSC-based models in FXS research, we do not deem necessary the inclusion of an extensive discussion of hESCs beyond what is currently presented in our introduction. We discuss two studies that have used hESCs as examples: one early paper that characterized features of FXS-hESCs and one that used KO hESCs. A separate review of hESC-based models in FXS should provide an in-depth discussion of published works on the topic. Moreover, we are concerned about potentially introducing bias to our cited references as the two specifically listed by the reviewer are by the same group.